# Quantum Data Assimilation:
# A New Approach to Solve Data Assimilation on Quantum Annealers

Shunji Kotsuki[1,2,3], Fumitoshi Kawasaki[4], and Masanao Ohashi[2]

[1]Institute for Advanced Academic Research, Chiba University, Chiba, Japan
[2]Center for Environmental Remote Sensing, Chiba University, Chiba, Japan
[3]Research Institute of Disaster Medicine, Chiba University, Chiba, Japan
[4]Graduate School of Science and Engineering, Chiba University, Chiba, Japan

*Correspondence to*: Shunji Kotsuki (shunji.kotsuki@chiba-u.jp)

**Abstract.**

Data assimilation is a crucial component in the Earth science field, enabling the integration of observation data with numerical models. In the context of numerical weather prediction (NWP), data assimilation is particularly vital for improving initial conditions and subsequent predictions. However, the computational demands imposed by conventional approaches, which employ iterative processes to minimize cost functions, pose notable challenges in computational time. The emergence of quantum computing provides promising opportunities to address these computation challenges by harnessing the inherent parallelism and optimization capabilities of quantum annealing machines.

In this investigation, we propose a novel approach termed quantum data assimilation, which solves data assimilation problem on quantum annealers. Our data assimilation experiments using the 40-variable Lorenz model were highly promising, showing that the quantum annealers produced analysis with comparable accuracy to conventional data assimilation approaches. In particular, the D-Wave System's physical quantum annealing machine achieved a significant reduction in execution time.

# 1 Introduction

Data assimilation is a mathematical discipline that integrates numerical models and observations data to improve the interpretation and predictions of dynamical systems (Reichle 2008; Evensen 2009). In particular, data assimilation has been intensively investigated in numerical weather prediction (NWP) during the past two decades to provide optimal initial conditions by combining model forecasts and observation data (Kalnay 2003; Houtekamer and Zhang 2016). Among data assimilation methods, variational and ensemble-variational data assimilation methods, which iteratively reduce cost functions
via gradient-based optimization, is widely used in most operational NWP centers such as the European Centre for Medium-Range Weather Forecasts (ECMWF), the United Kingdom Met Office (Met Office), the National Oceanic, Atmospheric Administration of the United States (NOAA) and Japan Meteorological Agency (JMA). However, the data assimilation methods require vast computational resources in NWP systems because of the iterations needed for sufficient cost function reduction. For example, in JMA's global forecast system, data assimilation requires about 25 times more computational
resources than forecast computations.

       In recent years, quantum computing has attracted research interest as a new paradigm of computational technologies since it has a large potential to overcome computational challenges of conventional approaches through quantum effects such as tunneling, superposition and entanglement. In particular, quantum annealing machines (Kadowaki and Nishimori 1998), such as D-Wave System's quantum annealers (Johnson et al. 2011), are powerful and feasible tools for solving optimization
problems. Since the quantum annealer 2000Q was released from D-Wave Systems in 2017, quantum computing research has rapidly progressed in various applications such as for machine learning (Wang et al. 2019; Willsch et al. 2020), graph partitioning (Ushijima-Mwesigwa et al. 2017), clustering (O'Malley et al. 2018), and model predictive control (Inoue et al. 2021).

       In this study, we design a data assimilation method for the quantum annealing machines. Although quantum machines
have been used in several engineering applications, to our knowledge, this is the first study to apply quantum annealing to data assimilation problems. We focus on the four-dimensional variational data assimilation (4DVAR) since it is the most widely-used data assimilation method in operational NWP systems. We reformulate the 4DVAR into the quadratic unconstrained binary optimization (QUBO) problem which can be solved by quantum annealers. We subsequently apply the proposed method for a series of 4DVAR experiments using a low-dimensional chaotic Lorenz 96 model (Lorenz 1996; Lorenz and Emanuel
1998), which has been widely used in theoretical data assimilation studies (e.g., Anderson, 2001; Whitaker and Hamill, 2002; Miyoshi, 2011; Kotsuki et al., 2017).

       The original 4DVAR cost function is, as elaborated below in section 2.1, a quadratic unconstrained optimization (QUO) problem including nonlinear operator with respect to the analysis increment (NL-QUO). To solve 4DVAR using quantum annealers, we first approximate the problem so as to include only linear operations with respect to the analysis (L-
QUO), which is then reformulated to a be quadratic unconstrained binary optimization (L-QUBO) problem. The L-QUBO problem is solved using D-Wave Advantage physical quantum annealer (Phy-QA), and the Fixstars Amplify's simulated

quantum annealer (Sim-QA). We also employ the conventionally-used quasi-Newton method with Broyden-Fletcher-Goldfarb-Shanno formula (BFGS) to solve the NL-QUO and L-QUO, which are denoted as NL-BFGS and L-BFGS hereafter. Numerical techniques and practical implementations specifically tailored to quantum data assimilation are also presented.

The rest of paper is organized as follows. Section 2 provides the method of quantum data assimilation, and Section 3 provides results and discussion. Finally, a summary is presented in section 4.

## 2 Methodology and experiments

### 2.1 Conventional data assimilation

This study focuses on the four-dimensional variational data assimilation (4DVAR), which is among the most widely used data assimilation methods in operational NWP centers such as ECMWF, Met Office, NOAA and JMA. The 4DVAR assimilates observations over a time window to produce an analysis trajectory that minimizes its cost function (Figure 1). The cost function of the 4DVAR is derived from Bayes' theorem, and is given by:

$$J(\delta\mathbf{x}_0) = \delta\mathbf{x}_0^T \mathbf{Q}_0^{-1} \delta\mathbf{x}_0 + \mathbf{d}_{1:L}^T \mathbf{R}_{1:L}^{-1} \mathbf{d}_{1:L}, \tag{1}$$

where $\delta\mathbf{x}_0 = \mathbf{x}_0 - \mathbf{x}_0^f$ ($\in \mathbb{R}^N$) is the analysis increment, $\mathbf{Q}$ ($\in \mathbb{R}^{N \times N}$) is the background error covariance, $\mathbf{d}_{1:L}$ ($\in \mathbb{R}^{PL}$) is the observation departure, and $\mathbf{R}_{1:L}$ ($\in \mathbb{R}^{PL \times PL}$) is the observation error covariance. The superscript $f$ represents the forecast, The subscripts 0 denotes at time $t = 0$ of the time window, and 1:$L$ indicates observation time slots from $t = 1$ to $t = L$. Here, $N$ is the system size, $L$ is the number of observation time slots, and $P$ is the number of observations per time slot. The observation departure is given by:

$$\mathbf{d}_{1:L}^T = [\mathbf{d}_1^T, \ldots, \mathbf{d}_k^T, \ldots, \mathbf{d}_L^T], \tag{2}$$

$$\mathbf{d}_k = \mathbf{y}_k^o - \mathbf{H}M_{k|0}(\mathbf{x}_0^f + \delta\mathbf{x}_0), \tag{3}$$

where $\mathbf{y}_k^o$ ($\in \mathbb{R}^P$) is the observation at time $k$, and $M_{k|0}()$ is the nonlinear model forecast from time $t = 0$ to $t = k$, $\mathbf{H}$ ($\in \mathbb{R}^{P \times N}$) is the linear observation operator. The superscript $o$ denotes the observation. The conventional 4DVAR iteratively updates the analysis increment $\delta\mathbf{x}_0$ by reducing the cost function using the quasi-Newton method based on the gradient:

$$dJ(\delta\mathbf{x}_0)/d\delta\mathbf{x}_0 = 2\mathbf{Q}_0^{-1}\delta\mathbf{x}_0 - 2\mathbf{M}_{1:L}^T\mathbf{H}_{1:L}^T\mathbf{R}_{1:L}^{-1}\mathbf{d}_{1:L}, \tag{4}$$

where

$$\mathbf{H}_{1:L}^T = \begin{bmatrix} \mathbf{H}^T & & \mathbf{O} \\ & \ddots & \\ \mathbf{O} & & \mathbf{H}^T \end{bmatrix}, \tag{5}$$

where $\mathbf{O}$ is the zero matrix. The adjoint model $\mathbf{M}_{1:L}^T$ ($\in \mathbb{R}^{N \times NL}$) is given by:

$$\mathbf{M}_{1:L|0}^T = \left[\mathbf{M}_{1|0}^T, \dots, \mathbf{M}_{k|0}^T, \dots, \mathbf{M}_{L|0}^T\right], \tag{6}$$

$$\mathbf{M}_{k|0}^T = \prod_{l=1}^{k} \mathbf{M}_{l|l-1}^T, \tag{7}$$

where $\mathbf{M}_{l|l-1}$ ($\in \mathbb{R}^{N \times N}$) is the tangent liner model that approximates:

$$M_{l|l-1}\left(M_{l-1|0}\left(\mathbf{x}_0^f + \delta\mathbf{x}_0\right) + \boldsymbol{\varepsilon}\right) \simeq M_{l|0}\left(\mathbf{x}_0^f + \delta\mathbf{x}_0\right) + \mathbf{M}_{l|l-1}\boldsymbol{\varepsilon}, \tag{8}$$

where $\boldsymbol{\varepsilon}$ ($\in \mathbb{R}^N$) is the vector with very small real numbers, and $\boldsymbol{\varepsilon} = 10^{-5} \times [1, \cdots, 1]^T$ is used in this study. This optimization minimizes the quadratic cost function (Eq. 1) including the nonlinear operator with respect to the analysis increment (Eq. 3), and referred as NL-QUO in this study. In NL-QUO, the tangent linear model $\mathbf{M}$ and its adjoint model $\mathbf{M}^T$ are updated at every iteration based on the latest analysis $\mathbf{x}_0 = \mathbf{x}_0^f + \delta\mathbf{x}_0$.

As an intermediate step toward QUBO, the original cost function is approximated as follows:

$$J(\delta\mathbf{x}_0) \simeq \tilde{J}(\delta\mathbf{x}_0) = \delta\mathbf{x}_0^T \mathbf{Q}_0^{-1} \delta\mathbf{x}_0 + \tilde{\mathbf{d}}_{1:L}^T \mathbf{R}_{1:L}^{-1} \tilde{\mathbf{d}}_{1:L}, \tag{9}$$

where

$$\tilde{\mathbf{d}}_{1:L}^T = \left[\tilde{\mathbf{d}}_1^T, \dots, \tilde{\mathbf{d}}_k^T, \dots, \tilde{\mathbf{d}}_L^T\right], \tag{10}$$

$$\mathbf{d}_k \simeq \tilde{\mathbf{d}}_k = \mathbf{y}_k^o - \mathbf{H}\mathbf{x}_k^f - \mathbf{H}\widetilde{\mathbf{M}}_{k|0}\delta\mathbf{x}_0, \tag{11}$$

$$\widetilde{\mathbf{M}}_{k|0} = \prod_{l=0}^{k-1} \widetilde{\mathbf{M}}_{l|l-1}. \tag{12}$$

Here, $\mathbf{x}_k^f = M_{k|0}\left(\mathbf{x}_0^f\right)$, and tilde indicates linear approximations. Unlike NL-QUO, this optimization retains the same tangent linear model and its adjoint model during the iteration. Namely, the tangent linear model is expanded not around the trajectory from the latest analysis ($\mathbf{x}_0^f + \delta\mathbf{x}_0$), but around the trajectory from the first guess ($\mathbf{x}_0^f$) as follows:

$$M_{l|l-1}\left(M_{l-1|0}\left(\mathbf{x}_0^f\right) + \boldsymbol{\varepsilon}\right) \simeq M_{l|0}\left(\mathbf{x}_0^f\right) + \widetilde{\mathbf{M}}_{l|l-1}\boldsymbol{\varepsilon}. \tag{13}$$

Substitution of Eqs. (10-12) into Eq. (9) gives the following quadratic unconstraint optimization (L-QUO) that has only linear operations with respect to the analysis increment as follows:

$$\tilde{J}(\delta\mathbf{x}_0) = \delta\mathbf{x}_0^T \left(\mathbf{Q}_0^{-1} + \widetilde{\mathbf{M}}_{1:L|0}^T \mathbf{H}_{1:L}^T \mathbf{R}_{1:L}^{-1} \mathbf{H}_{1:L} \widetilde{\mathbf{M}}_{1:L|0}\right)\delta\mathbf{x}_0 - 2\mathbf{s}_{1:L}^T \mathbf{R}_{1:L}^{-1} \mathbf{H}_{1:L} \widetilde{\mathbf{M}}_{1:L|0}\delta\mathbf{x}_0 + C, \tag{14}$$

where the constant $C$ is:

$$C = \mathbf{s}_{1:L}^T \mathbf{R}_{1:L}^{-1} \mathbf{s}_{1:L}, \tag{15}$$

$$\mathbf{s}_{1:L}^T = \left[\mathbf{s}_1^T, \dots, \mathbf{s}_k^T, \dots, \mathbf{s}_L^T\right], \tag{16}$$

$$\mathbf{s}_k = \mathbf{y}_k^o - \mathbf{H}\mathbf{x}_k^f. \tag{17}$$

The gradient of $\tilde{J}$ is given by:

$$\frac{d\tilde{J}(\delta\mathbf{x}_0)}{d\delta\mathbf{x}_0} = 2\left(\mathbf{Q}_0^{-1} + \widetilde{\mathbf{M}}_{1:L|0}^T\mathbf{H}_{1:L}^T\mathbf{R}_{1:L}^{-1}\mathbf{H}_{1:L}\widetilde{\mathbf{M}}_{1:L|0}\right)\delta\mathbf{x}_0 - 2\widetilde{\mathbf{M}}_{1:L|0}^T\mathbf{H}_{1:L}^T\mathbf{R}_{1:L}^{-1}\mathbf{s}_{1:L}. \tag{18}$$

## 2.2 Quantum data assimilation

Quantum annealers require only the cost function in contrast to conventional 4DVAR that require the cost function and its gradient. However, the cost function should be represented by binary variables (i.e., 0 or 1) for quantum annealers. In this study, we represented a real number with $Z$ qubits where $Z$ is a natural number. In accordance with the method of Inoue et al. (2021), a mapping matrix $\mathbf{G}$ ($\in \mathbb{R}^{N \times NZ}$) is used in this study:

$$\delta\mathbf{x}_0 \simeq \frac{1}{\alpha}\mathbf{Gb} = \frac{1}{\alpha}\begin{bmatrix} \mathbf{g}^T & & \mathbf{0} \\ & \ddots & \\ \mathbf{0} & & \mathbf{g}^T \end{bmatrix}\mathbf{b}, \tag{19}$$

$$\mathbf{g}^T = [-2^{Z-1}, 2^{Z-2}, 2^{Z-3}, \dots, 2^1, 2^0], \tag{20}$$

where $\alpha$ is the tunable scaling parameter, $\mathbf{b}$ ($\in \mathbb{R}^{NZ}$) is the binary vector whose elements are all either 0 or 1, and $\mathbf{0}$ is the vector whose elements are all 0. Thus, Eq. (14) is reformulated into QUBO as follows:

$$\tilde{J}(\delta\mathbf{x}_0) \simeq \mathcal{H}(\mathbf{b}) = \mathbf{b}^T\mathbf{Ab} + \mathbf{u}^T\mathbf{b} + C, \tag{21}$$

where $\mathcal{H}$ is the Hamiltonian, and $\mathbf{A}$ ($\in \mathbb{R}^{NZ \times NZ}$) and $\mathbf{u}$ ($\in \mathbb{R}^{NZ}$) are given as follows:

$$\mathbf{A} = \frac{1}{\alpha^2}\mathbf{G}^T\left(\mathbf{Q}_0^{-1} + \widetilde{\mathbf{M}}_{1:L|0}^T\mathbf{H}_{1:L}^T\mathbf{R}_{1:L}^{-1}\mathbf{H}_{1:L}\widetilde{\mathbf{M}}_{1:L|0}\right)\mathbf{G}, \tag{22}$$

$$\mathbf{u}^T = -\frac{2}{\alpha}\mathbf{s}_{1:L}^T\mathbf{R}_{1:L}^{-1}\mathbf{H}_{1:L}\widetilde{\mathbf{M}}_{1:L|0}\mathbf{G}. \tag{23}$$

Note that the constant $C$ is irrelevant to the minimization problem. Quantum annealers solve the QUBO problem (Eq. 21) by inputting $\mathbf{A}$ and $\mathbf{u}$ to the solvers.

## 2.3 Experiments with the Lorenz-96 model

This study performs twin idealized experiments using the 40-variable Lorenz-96 model, which has been used widely in theoretical data assimilation studies (e.g., Anderson, 2001; Whitaker and Hamill, 2002; Miyoshi, 2011; Kotsuki et al., 2017). The Lorenz-96 model is defined as follows:

$$\frac{dx_i}{dt} = (x_{i+1} - x_{i-2})x_{i-1} - x_i + F, \quad (i = 1,2,\dots,N), \tag{24}$$

where the boundary is cyclic (i.e., $x_i = x_{i+N} = x_{i-N}$), and the model dimension $N$ is 40. The forcing is fixed at $F = 8.0$, which makes the model behave chaotically. The model is integrated using a fourth-order Runge–Kutta scheme with a non-dimensional time step of 0.05. A time step of 0.05 is considered to be 6 hours following Lorenz and Emanuel (1998). This study uses the identity matrix for the observation error covariance, such that variance of the uncorrelated observation error was 1.0. We first employed the nature run by running the Lorenz-96 model, and observation data were generated around the nature run every 6 hours (0.05 time units). All grid points are observed. The background error covariance is tuned to be $\mathbf{Q} = 0.15\mathbf{I}$.

As in conventional 4DVAR, NL-QUO and L-QUO are solved using the quasi-Newton method with Broyden-Fletcher-Goldfarb-Shanno (BFGS) formula (Table 1). In our BFGS-based 4DVAR experiments, we use the AMD's Ryzen 7 5800U with a Radeon Graphics as the CPU. All source codes are written in the Python v3.8.10. The NumPy and SciPy libraries are used for numerical computations. We first conducted NL-QUO 4DVAR data assimilation cycles 50 times over a period of 100 days, with a 2-day time window. Then, we preserved the 50 first guess data which are used for the other three approaches subsequently: L-QUO by BFGS, and L-QUBO by simulated and physical quantum annealers. Namely, our experiments purely compare the optimization processes of data assimilation among the four approaches.

## 2.4 Quantum annealers

As the experimental environment of the Phy-QA, we use D-Wave Advantage System 4.1 which consists of 5,627 physical qubits and 177 logical qubits, respectively (D-Wave, 2022). Because the D-Wave Advantage performs computations using the logical qubits, the 177 logical qubits are available for computations. Because of this limitation, we represent a real number (an analysis increment for one variable of the Lorenz-96 model) with four logical qubits ($Z = 4$) to limit the size of binary vector $\mathbf{b}$ less than 177 logical qubits (i.e., $N \times Z = 40 \times 4 = 160 \leq 177$). The computational time of the D-Wave's optimizations are measured using the "execution_time" function. Quantum annealers gives $\mathbf{b}$ as a solution of QUBO problem by submitting input parameters $\mathbf{A}$ and $\mathbf{u}$ to the solvers (Eq. 21). Here, it is essential to emphasize that physical quantum annealers do not employ traditional algorithms to solve QUBO problems; instead, they obtain solutions $\mathbf{b}$ as a result of quantum effects. Additional details on quantum annealers can be found in Appendix A.

For simulated quantum annealer, we use the Fixstars Amplify's simulated quantum annealer (Fixstars Amplify Annealing Engine) which incorporates $\geq 65,536$ physical and logical qubits (Matsuda 2020). The simulated quantum annealer emulates quantum effects via digital computations using graphics processing unit (GPU). Although the simulated quantum annealers cannot cause real quantum effects, it is applicable to large-scale calculations due to larger logical qubits than the Phy-QA. The computational time is also measured by the "execution_time" function for Sim-QA. Despite the larger number of logical qubits available in the Fixstars Amplify's Sim-QA, we use four logical qubits for a real number ($Z = 4$) in numerical experiments to compare with the D-Wave's Phy-QA. Consequently, we use 160 logical qubits for both of Sim-QA and Phy-QA experiments.

## 3 Results and discussion

### 3.1 Performance of quantum data assimilation

Figures 2 (a) and (b) show a comparison of the mean analysis and 2-day forecast root mean square errors (RMSEs) w.r.t. the nature run for the four different approaches. As anticipated, the NL-BFGS achieved the lower RMSEs as it did not approximate the original cost function. Approximating the nonlinear operations of the original cost function led to a slight increase in the analysis and forecast RMSEs as observed in L-BFGS. Solving QUBO using the quantum annealers resulted in

reduced analysis errors compared to the first guess, as demonstrated by the Sim-QA and Phy-QA. Here, tunable parameters for quantum annealers, namely the scaling factor $\alpha$ and "num_reads," were calibrated prior to experiments, and will be discussed in the subsequent subsection (section 3.2). Although Phy-QA successfully reduced analysis errors when compared to the first guess, it exhibited a slightly larger analysis RMSE than the other three approaches. The larger RMSE of Phy-QA is intensified when we see the 2-day forecast RMSE. This discrepancy could be attributed to the stochastic quantum effects inherent in D-Wave's quantum annealer, as discussed in detail in the next subsection.

On the other hand, Phy-QA demonstrated the fastest execution time among the four approaches (Figure 2 c). Here, the NL-BFGS exhibited the longest computation time due to the iterative updates of the trajectory and its tangent linear and adjoint models. In contrast, L-BFGS, which retains the first-guess-based trajectory, was significantly faster than NL-BFGS. Sim-QA required a longer execution time than Phy-QA, presumably because GPU-based simulated quantum annealers involve computations for the artificial emulation of quantum effects. The D-Wave's Phy-QA obtained a significant reduction in computation time compared to the other three approaches (NL-BFGS, L-BFGS and Sim-QA), taking less than 0.05 s to find a solution.

It should be noted that there are slight differences in analysis and forecast RMSEs between NL-BFGS and L-BFGS in Figures 2 (a) and (b). The degradations of L-BFGS w.r.t. NL-BFGS would be more pronounced for stronger nonlinear cases, such as those with longer time windows of 4DVAR, since Eqs. (9)–(13) directly simplify the nonlinear operator in the 4DVAR to a linear operator. Here, our quantum data assimilation solves a QUBO problem (Eq. 21), which is an approximation of the cost function solved in L-BFGS (Eq. 14). Therefore, Sim-QA and Phy-QA would be worsened too for stronger nonlinear cases.

Figure 3 provides an arbitrary-selected data assimilation example, where the cost functions of NL-QUO and L-QUO are depicted by blue and red contour lines, respectively. Note that Figure 3 shows an example of analysis while Figure 2a provides the mean RMSEs averaged over 50 data assimilations. The BFGS-based 4DVAR (NL-BFGS and L-BFGS) gradually converged towards their respective analyses through iterative updates. Consequently, both NL-BFGS and L-BFGS yielded analyses that are closer to the minima of their respective cost functions compared to the first guess (blue and red triangles in Figure 3). In contrast, Sim-QA and Phy-QA produced a single analysis each, as they do not involve iterations. In this specific example, Sim-QA, aiming to minimize L-QUBO, generated an analysis (as a yellow circle) that was distant from the bottom of L-QUO. Conversely, the analysis produced by Phy-QA (a magenta circle) is closer to the minima of NL-QUO. Notably, despite solving the same L-QUBO problem, Sim-QA and Phy-QA yielded greatly different analyses in this example. This discrepancy is presumed to be a result of stochastic quantum effects, which will be further investigated in the next subsection.

## 3.2 Sensitivity to tunable parameters

It is important to mention that the D-Wave's physical quantum annealer produces non-deterministic outputs due to stochastic quantum effects. Therefore, the quantum annealer includes a tunable parameter called "num_reads," which defines the number of states (output solutions) to be read from the solver. Generally, a larger value of "num_reads" increases the probability of obtaining a better solution. Here, the better solution, which results in smaller Hamiltonian $\mathcal{H}$ (Eq. 21), is

200 expected to yield smaller analysis RMSEs. In this study, we investigated the sensitivity of Phy-QA to the "num_reads" parameter. Figure 4 illustrates the sensitivity of the analysis RMSE and the mean computational time with respect to the "num_reads" parameter. For each "num_reads" value, we repeated 50 data assimilations ten times. The mean RMSE and standard deviation over the ten samples are presented in Figure 4 (a). Increasing the value of "num_reads" generally resulted in a reduction of the analysis RMSE. While the execution time exhibited a linear increase with respect to "num_reads" (Figure

4 b), reading out solutions from the solver multiple times proved beneficial in achieving stable and improved analyses. Importantly, the standard deviation of the mean RMSE was found to be insensitive to the "num_reads" parameter. This observation indicates that even if we read out the output from the solver over 100 times, the stochastic effects persist and cannot be eliminated for Phy-QA.

The scaling factor $\alpha$ is an important parameter for regulating analysis accuracy in quantum annealers (cf. Eq. 19).
Due to the limited number of available logical qubits for Phy-QA, this study represented real numbers using only four logical qubits. In other words, both our Phy-QA and Sim-QA could only represent $2^4 = 16$ distinct real numbers. For instance, a scaling factor $\alpha = 20$ indicates that numerical experiments can handles real numbers ranging from $-0.40$ to $0.35$ with an increment of 0.05. Our sensitivity experiments to the scaling factor $\alpha$ revealed similar increasing trends for Phy-QA and Sim-QA when $\alpha \geq 50$ (Figure 5). Increasing the scaling factor $\alpha$ resulted in degraded analysis accuracy, approaching the RMSE
of the first guess. This is because excessively large scaling factors can lead to slight changes in the analysis increment. For instance, when $\alpha = 500$, the minimum and maximum analysis increments are -0.016 and 0.014, respectively. It indicates that when $\alpha = 500$, Phy-QA and Sim-QA can manage a limited increment (from -0.016 to 0.014); as a result, their RMSEs approach the RMSE of the first guess. Interestingly, noticeable discrepancies were observed between Phy-QA and Sim-QA for $\alpha \leq 20$, presumably due to the stochastic quantum effects specific to Phy-QA. As observed in Figure 4, we cannot
eliminate stochastic effects for Phy-QA even if we read out output from the solver over 100 times. For smaller scaling factors, a stochastic change in a single qubit can induce larger changes in the analysis increment. Consequently, the optimal scaling factor for Phy-QA was found to be larger than that for Sim-QA. Based on the results of the sensitivity experiments, scaling factors of $\alpha = 20$ and $\alpha = 50$ were used for Sim-QA and Phy-QA in section 3.1. These sensitivity experiments also indicate that the optimal scaling factor may differ between Sim-QA and Phy-QA.

**4 Summary**

This study proposed the quantum data assimilation which solves the data assimilation problems on quantum annealing machines. The main results of this investigation are briefly listed as follows:

1) We reformulated the data assimilation problem into the quadratic unconstrained binary optimization (QUBO) problem so that quantum annealing machines can solve the data assimilation.

2) Using the 40-variable Lorenz model, we succeeded in solving data assimilation on quantum annealers for the first time. The results of our experiments were highly promising, demonstrating that the quantum annealers can yield analysis whose accuracy is comparable to conventional the quasi-Newton-based iterative approach.

3) The D-Wave's physical quantum annealing machine needed execution time less than 0.05 second, which is significantly smaller than conventional approaches.

4) Since the D-Wave's physical quantum annealer produces non-deterministic outputs due to stochastic quantum effects, reading out solutions multiple times was beneficial in achieving stable and improved analyses.

5) The scaling factor for quantum data assimilation is an important parameter for regulating analysis accuracy in our configuration. Due to the stochastic quantum effects, the optimal scaling factor for D-Wave's physical quantum annealing machine was different from the simulated quantum annealer.

At the time of writing this manuscript, the number of logical qubits in the physical quantum annealer ($O(10^2)$) is far from meeting the large-scale computation requirements of practical NWP models ($> O(10^8)$). To extend our approach for high-dimensional models such as NWP models, employing dimensional reduction would be necessary and helpful. The dimensional reduction techniques have been used in data assimilation such as in operational 4DVARs that solve the inverse problem in spectral space (e.g. Bonavita et al. 2015). Furthermore, recent studies proposed solving data assimilation in low-dimensional latent space spanned by deep-learning-based nonlinear functions (e.g. Peyron et al. 2021). A multi-dimensional inverse problem can be converted into a unitless and normalized inverse problem using these dimensional reduction techniques. Therefore, these dimensional techniques are also beneficial to avoid tuning the scaling parameter $\alpha$ for each variable. Future research is necessary to explore how to integrate these dimensional reduction methods with quantum data assimilation.

We anticipate that our findings will inspire future developments in the application of quantum technologies to advance data assimilation to reach deeper understating and improved predictions of real-world complex systems in NWP and beyond. In addition, our work would advance the practical applications of quantum annealing machines in solving complex optimization problems in Earth science.

**Appendix A: Quantum Annealer**

This appendix describes how quantum annealers reach to solutions of the QUBO problems. First of all, it should be noted that there are two kinds of quantum computers at this moment: quantum annealing machines and quantum gate machines. The quantum gate machines are general-purpose quantum devices capable of performing a wide range of quantum computations. The quantum gate machines enable users to design quantum circuits which manipulate qubits through gate operations to solve complex problems efficiently.

The quantum annealers are devices specialized for solving optimization problems written by QUBO problem or Ising model. Here, a problem written by the Ising model can be reformulated to an equivalent QUBO problem mathematically, and vice versa. Figure A1 provides a conceptual image of the quantum annealing. Users can submit jobs (i.e., problems written by QUBO or Ising model) to quantum annealers through an application programming interface (API) (Figure A1 a). This study used the Fixstars Amplify's software development kit (known as Amplify SDK) as an API to submit jobs to D-Wave's physical quantum annealer and the Fixstars Amplify's simulated quantum annealer. Figure A1 (b) shows the functions of the control device, quantum processing unit (QPU), and measuring device, respectively. The control device regulates magnetic fields to

tune the Hamiltonian of the quantum system, guiding to a low-energy state corresponding to an optimal solution of QUBO. In QPU, quantum effects (superposition and entanglement) are leveraged to explore potential solutions of the QUBO problem through quantum annealing (Figure A1 c). The measuring device reads the final state of the QPU. Here, "num_reads" defines the number of states to be read from the QPU by the measuring device. Finally, a user can obtain a solution (the binary vector **b** in this study) that yielded the smallest Hamiltonian among the "num_reads" states read by the measuring device.

### Author contributions

S. Kotsuki developed methodology of the study, F. Kawasaki employed Lorenz-96 experiments, and M. Ohashi conducted experiments on quantum annealers.

### Code availability

The code that supports the findings of this study is available from the corresponding author upon reasonable request.

### Data availability

All of the data and codes used in this study are stored for 5 years at Chiba University, and are available from the corresponding author upon request by contacting the corresponding author.

### Competing interests

The authors have no competing interests to declare.

### Acknowledgements

This study was partly supported by the JST Moonshot R&D (JPMJMS2284, JPMJMS2389), JST PRESTO (JPMJPR1924), the Japan Society for the Promotion of Science (JSPS) KAKENHI grants JP21H04571, JP21H05002, JP22K18821, and the IAAR Research Support Program of Chiba University.

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

**Tables**

**Table 1**: Experiments conducted in this study. Outer loop indicates that the trajectory is updated during the 4DVAR. Inner loop indicates that the analysis increments are computed iteratively by the quasi-Newton method with BFGS formula.

| Experiments | Cost function | Solver | Outer loop | Inner loop |
|---|---|---|---|---|
| NL-BFGS | NL-QUO | Quasi-Newton with BFGS | × | × |
| L-BFGS | L-QUO | Quasi-Newton with BFGS | | × |
| Sim-QA | L-QUBO | Simulated QA | | |
| Phy-QA | L-QUBO | Physical QA | | |

**Figures**

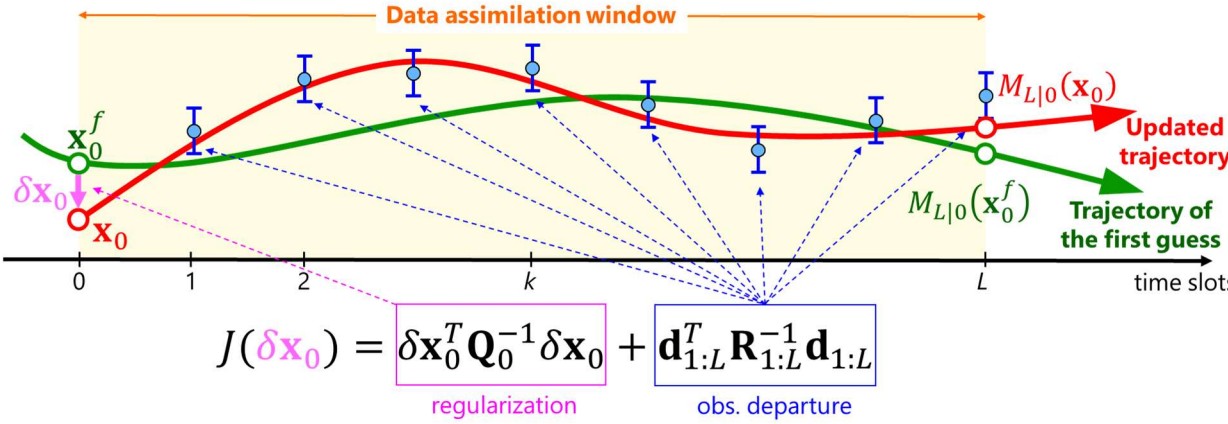

$$J(\delta\mathbf{x}_0) = \boxed{\delta\mathbf{x}_0^T \mathbf{Q}_0^{-1} \delta\mathbf{x}_0} + \boxed{\mathbf{d}_{1:L}^T \mathbf{R}_{1:L}^{-1} \mathbf{d}_{1:L}}$$

regularization   obs. departure


**Figure 1:** Conceptual image of 4DVAR data assimilation. Green and red lines indicate trajectories of the first guess and updated analysis, respectively. Blue circles with error bars represent observations.

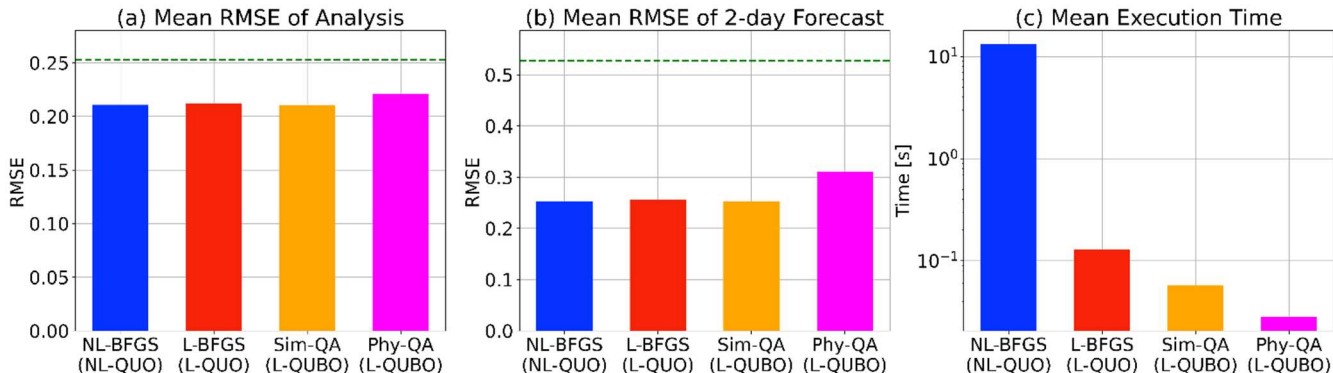

**Figure 2:** (a) Mean analysis root mean square errors (RMSEs) at the initial time of 4DVAR data assimilation window (a-RMSE), (b) mean 2-day forecast RMSEs (f-RMSE), and (c) mean execution time (s) over 50 data assimilations for (blue) NL-BFGS (a-RMSE = 0.2109; f-RMSE = 0.2520; 13.18 s), (red) L-BFGS (a-RMSE = 0.2120; f-RMSE = 0.2554; 0.129 s), (orange) Fixstars Amplify Sim-QA (a-RMSE = 0.2105; f-RMSE = 0.2531; 0.057 s), and (magenta) the D-Wave Phy-QA (a-RMSE = 0.2209; f-RMSE = 0.3101; 0.028 s). The green dashed line in (a) indicates the first-guess prior to the data assimilation (a-RMSE = 0.2530; f-RMSE = 0.5267). The "num_reads" of Phy-QA was set at 50. The scaling factor $\alpha$ was set at 20 and 50 for Sim-QA and Phy-QA, respectively.

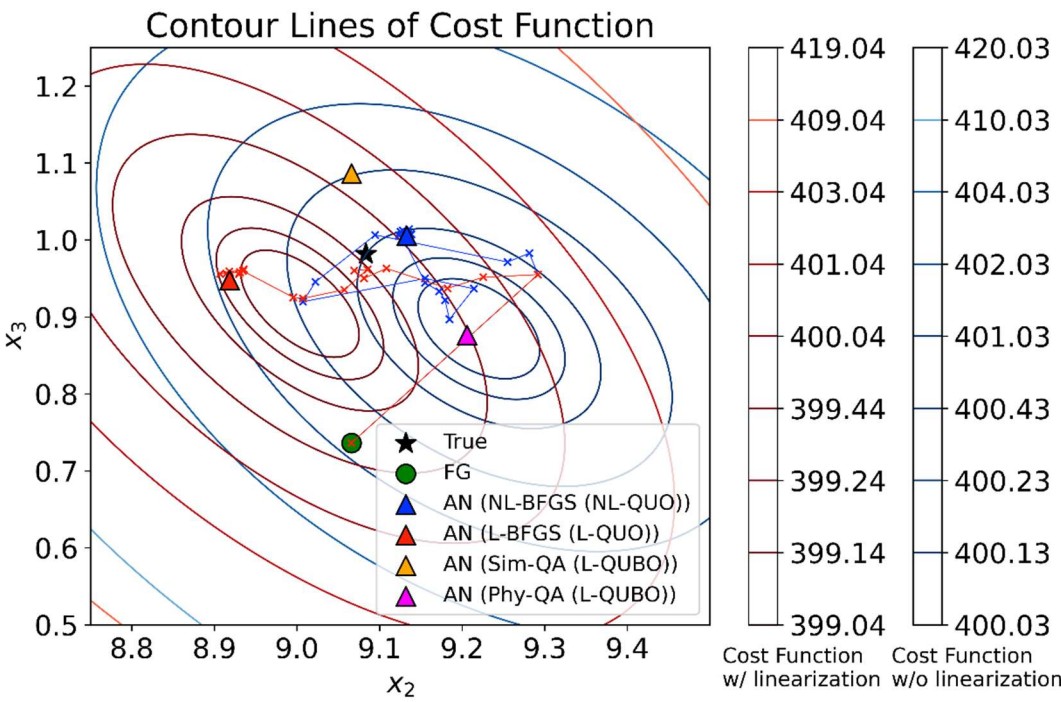


**Figure 3:** An illustration of data assimilations from an arbitrary-selected first guess. The black star and green circle indicate the truth and the first guess, respectively. Blue, red, orange, and magenta triangles are the analyses of NL-BFGS, L-BFGS, Sim-QA, and Phy-QA, respectively. Blue and red lines represent the analysis updates over iterations for NL-BFGS and L-BFGS whose internal analyses are indicated by the cross marks. Red and blue contours show the cost functions with and

without the linearization (L-QUO and NL-QUO), respectively. These cost functions were computed in a grid search for two variables ($x_2$ and $x_3$) of the Lorenz 96 model, with true data used for the other 38 variables. Since the BFGS-based 4DVAR algorithm updates 40 variables, analyses of NL-QUO and L-QUO (blue and red circles) are not exactly placed at the minima of their cost functions. The "num_reads" of Phy-QA was set at 50. The scaling factor α was set at 20 and 50 for Sim-QA and Phy-QA, respectively.


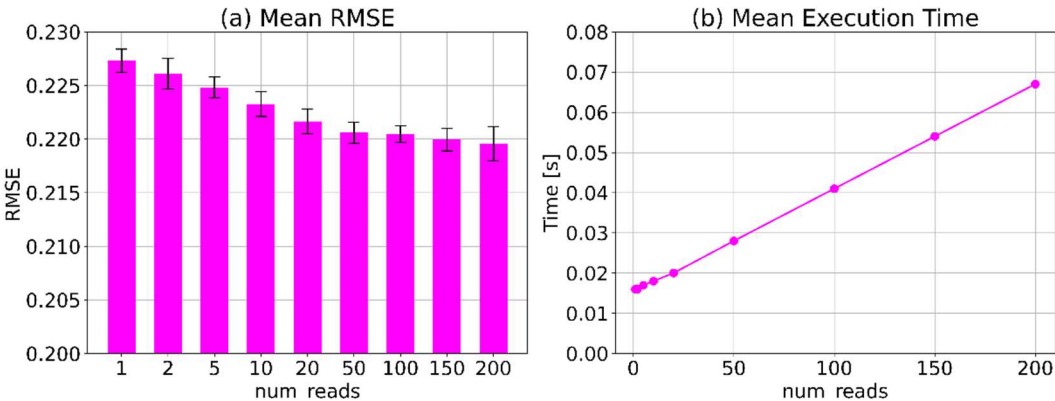

**Figure 4:** Sensitivity of (a) analysis RMSE and (b) mean execution time to the tunable parameter "num_reads" in D-Wave Phy-QA. We conducted 50 data assimilations 10 times, and their means and standard deviations over 10 samples are represented by bars and error bars. The scaling factor $\alpha$ is set at 50.

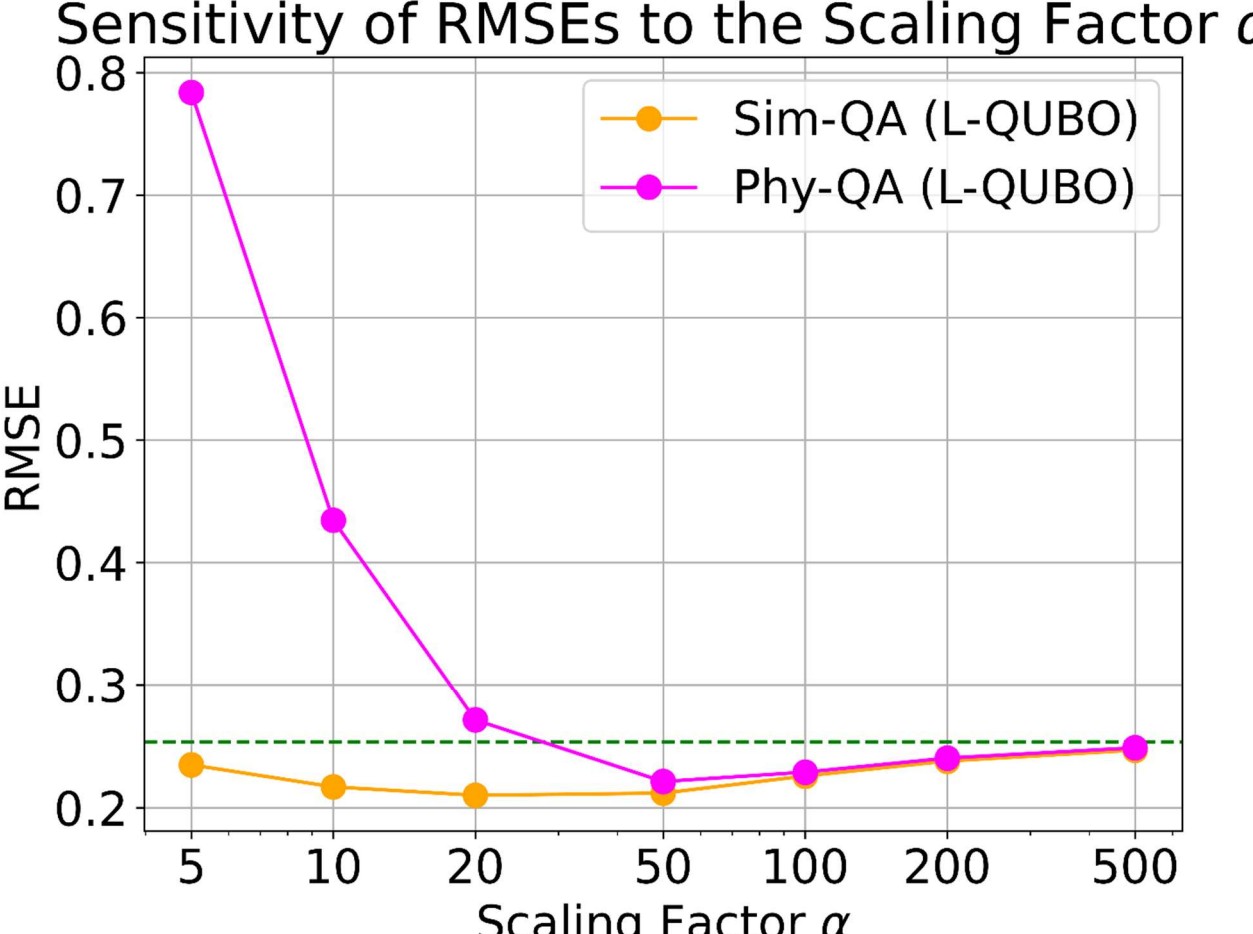


**Figure 5:** Sensitivity of analysis RMSEs to the scaling factor $\alpha$ for Sim-QA (orange), and Phy-QA (magenta). The green-colored dashed line in (a) represents the first-guess RMSE prior to the data assimilation. The parameter "num_reads" of Phy-QA is set at 50.

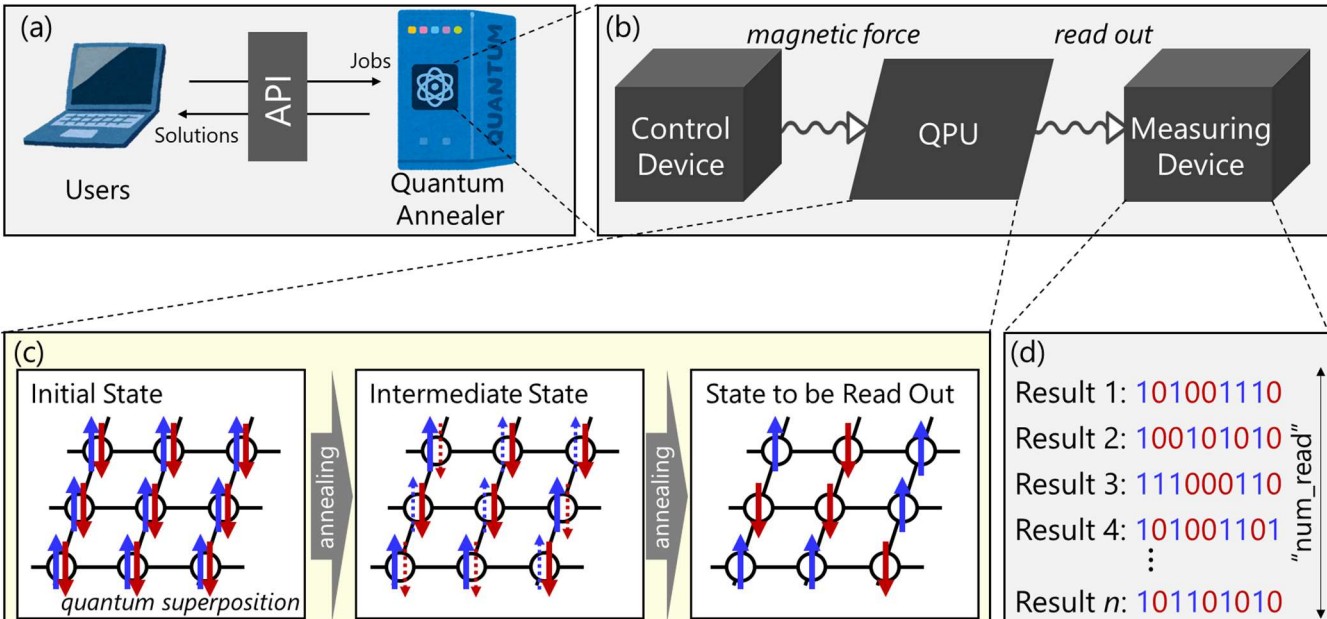

**Figure A1:** A conceptual image of the quantum annealing. (a) How users submit jobs to quantum annealers through an application programming interface (API). (b) Schematic image of the control device, quantum processing unit (QPU), and measuring device in the D-Wave's quantum annealers. (c) Schematic image of quantum annearling from an initial state with quantum superposition to a final state to be read out. Blue and red arrows in (c) represent the upward and downward spins of qubits, respectively. (d) Schematic image of states that are read out by the measuring device. Here, "num_read" indicates the number of times the quantum annealing process is executed and the resulting states are read out. Finally, the best result, which yielded the smallest Hamiltonian among the "num_reads" results, is returned to users.