# Peer review of "Quantum Data Assimilation"

_Nonlinear Processes in Geophysics, 2023_

## Referee Comment (RC2)

Review of "Quantum Data Assimilation: A New Approach to Solve Data Assimilation on Quantum Annealers" by S. Kotsuki, F. Kawasaki, and M. Ohashi.

**General comments:**

This manuscript proposes a novel approach named quantum data assimilation and claims that execution time is reduced significantly with the D-Wave System's physical quantum annealing machine. In my opinion, this work is valuable, combining data assimilation and quantum computation. However, several problems prevent me from recommending the manuscript for publication in its present form.

**Specific comments:**

(1) In this manuscript, the introduction to data assimilation methods and quantum computation, especially the quantum annealers, is a little brief. Some essential references should be also included.

(2) In line 54 of the manuscript, the authors explain briefly how to solve the 4D-VAR using the quantum annealers. The authors should elucidate the algorithm in the method section.

(3) To my knowledge, in quantum computation, quantum states are operated, like $|0\rangle$, $|1\rangle$, $\alpha|1\rangle + \beta|0\rangle$, nor binary variables (0 or 1). So, actually, the algorithm proposed in this work may be not relevant to the quantum computation. The authors need to explain this query clearly.

(4) Can the proposed method be used for real numerical models (for example the WRF model) and what are the operational difficulties?

(5) How does the method affect the accuracy of numerical forecasts in the design of numerical experiments? Please provide additional experiments.

---

## Author Comment (AC1)

**[Response to Reviewer's Comments]**

We are very grateful to the reviewers and Dr. Banglin Zhang for her/his careful reviews and kindly giving us valuable and constructive comments and suggestions that we have generally accepted. Here, we provide our point-by-point responses whose P and L correspond to page and line numbers of the supplemental PDF file. The revisions are highlighted by red in the revised manuscript. Supplemental PDF file would be useful to check revisions and their corresponding comments

======================================================================

**[Reviewer 1 General Comments]**

In this study, the authors propose using the Quantum Annealers to solve the data assimilation in the Earth science field. This is the first study to employ quantum annealing for data assimilation problems. The quantum annealing machine can significantly reduce the execution time, which will be very useful for the development of the data assimilation. However, there are several problems that need to be clarified in the current version of the manuscript.

Response: Thank you for the comment. We revised the manuscript following comments.

**[Reviewer 1 Specific Comments]**

(1) In the data assimilation field, many readers may be not familiar with the quantum optimization. So could you give more detailed statements about the quantum optimization algorithm? Maybe, you can add an appendix for the quantum optimization algorithm and how to use it.

Response: We added explanations on this point (P6L149, Appendix A).

(2) In this manuscript, the quantum data assimilation was only tested using the 40-variable Lorenz-96 model. This is a relatively simple model. In the realistic atmosphere and ocean data assimilation, the model is complicate. In this situation, how to carry out the quantum data assimilation? Especially, how to select the values of parameters in Eq. (19)? Can you also discuss the possible advantages and disadvantages of the quantum data assimilation in the realistic application?

Response: We added discussion on this point (P9L240).

(3) Figure 3 shows that the NL-BFGS is closer to the truth than the Sim-QA. This seems to be inconsistent with Fig. 2a. Why?

Response: We added explanation on this point (P7L186).

(4) To conduct the quantum data assimilation, the linearization needs to be done. For a strong nonlinear case, whether quantum data assimilation cannot be used?

Response: We also added discussion on this point (P7L180).

(5) Some details need to be clarified. For example, in the Lorenz-96 model, how to generate the observation data? How to perform the control run?

Response: We added explanations on this point (P6L134).

(6) Line 114, Inoue and Yoshida (2021) should be Inoue et al. (2021)?

Response: Revised (P5L114).

================================================================================

**[Reviewer 2 General Comments]**

This manuscript proposes a novel approach named quantum data assimilation and claims that execution time is reduced significantly with the D-Wave System's physical quantum annealing machine. In my opinion, this work is valuable, combining data assimilation and quantum computation. However, several problems prevent me from recommending the manuscript for publication in its present form.

Response: Thank you for encouraging comment. We revised the manuscript accordingly following suggestions.

**[Reviewer 2 Specific Comments]**

(1) In this manuscript, the introduction to data assimilation methods and quantum computation, especially the quantum annealers, is a little brief. Some essential references should be also included.

(2) In line 54 of the manuscript, the authors explain briefly how to solve the 4D-VAR using the quantum annealers. The authors should elucidate the algorithm in the method section.

(3) To my knowledge, in quantum computation, quantum states are operated, like , ,, nor binary variables (0 or 1). So, actually, the algorithm proposed in this work may be not relevant to the quantum computation. The authors need to explain this query clearly.

Response: We added explanations on this point (P6L149, Appendix A) to answer for your specific comments (1)-(3).

(4) Can the proposed method be used for real numerical models (for example the WRF model) and what are the operational difficulties?

Response: We added discussion on this point (P9L240).

(5) How does the method affect the accuracy of numerical forecasts in the design of numerical experiments? Please provide additional experiments.

Response: We added additional forecast experiments in the revised manuscript (Figure 2 b and P6L163).

======================================================================

**[General Comments by Dr. Banglin Zhang]**

This paper applied the quantum annealing method to improve the computational efficiency of the four-dimensional variational assimilation (4D-Var) minimization process, resulting in significant results and great potential for practical applications.

Response: Thank you very much for taking you time to review our manuscript. We revised the manuscript following your comments.

**[Specific Comments by Dr. Banglin Zhang]**

(1) When comparing data assimilation methods, it is necessary to not only consider the accuracy of the initial field but also its impact on the forecast error of the model. However, this paper only compared the accuracy of the analysis field and does not demonstrate its impact on the forecast error of the model.

Response: We added additional forecast experiments in the revised manuscript (Figure 2 b and P6L163).

(2) Equations (9)-(13) directly simplify the nonlinear operator in the 4D-Var to a linear operator, which should lead to a noticeable decrease in the accuracy of the analysis field (as it is no longer a tangent linear approximation). However, in Figure 2, there seems to be no difference in RMSE between NL-BFGS and L-BFGS. What could be the reason for this?

Response: We added discussion on this point (P7L180).

(3) The number of physical and logical nodes in Phy-QA (5627) is much smaller than in Sim-QA (≥65,536). Would this have an impact on the comparison of computational efficiency later?

Response: We added an explanation on this point (P6L159).

(4) In line 171, "three" should be "two".

Response: We added an explanation on this point (P7L178).

(5) In line 306, "Red and blue" should be "Blue and red".

Response: Revised (Caption of Figure 5).

(6) In the first paragraph of section 3.2, since increasing "num_reads" does not reduce the randomness of Phy-QA, how does it reduce the RMSE of the analysis field?

Response: We added explanation on this point (P7L199).

(7) In lines 198-200, I don't understand why a change in the maximum and minimum analysis increments when α=500 would result in decreased accuracy of the analysis field.

Response: We added description on this point (P8L216).

(8) It seems that the number of logical nodes in the quantum annealer is far from meeting the large-scale computation requirements of practical weather forecasting models.

Response: We added discussion on this point (P9L240).